# Depth Any Video with Scalable Synthetic Data

**Honghui Yang**[1,2*], **Di Huang**[4*], **Wei Yin**[2], **Chunhua Shen**[1], **Haifeng Liu**[1†]
**Xiaofei He**[1], **Binbin Lin**[3], **Wanli Ouyang**[2], **Tong He**[2†]

[1]State Key Lab of CAD&CG, Zhejiang University     [2]Shanghai AI Laboratory
[3]School of Software Technology, Zhejiang University     [4]The University of Sydney

`https://depthanyvideo.github.io`

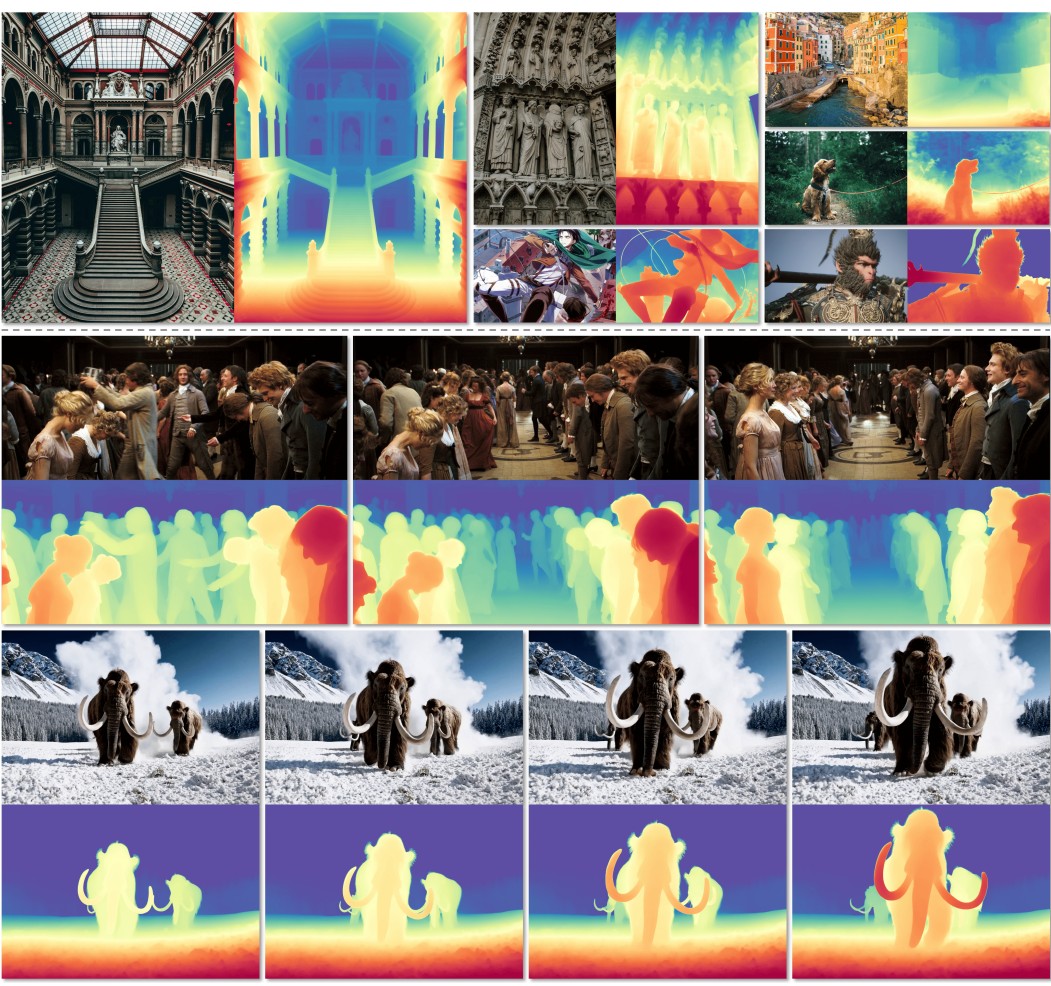

Figure 1: We present **Depth Any Video**, a versatile foundation model supporting both image (top half) and video (bottom half) depth estimation. Derived from Stable Video Diffusion and fine-tuned with diverse and high-quality synthetic data, our model achieves remarkably robust generalization across various real and synthetic unseen scenarios. Additionally, it faithfully captures intricate fine-grained details while ensuring temporal consistency throughout the video.

## Abstract

Video depth estimation has long been hindered by the scarcity of consistent and scalable ground truth data, leading to inconsistent and unreliable results. In this paper, we introduce **Depth Any Video**, a model that tackles the challenge through two key innovations. First, we develop a scalable synthetic data pipeline, capturing real-time video depth data from diverse virtual environments, yielding 40,000 video clips of 5-second duration, each with precise depth annotations. Second, we leverage the powerful priors of generative video diffusion models to handle real-

world videos effectively, integrating advanced techniques such as rotary position encoding and flow matching to further enhance flexibility and efficiency. Unlike previous models, which are limited to fixed-length video sequences, our approach introduces a novel mixed-duration training strategy that handles videos of varying lengths and performs robustly across different frame rates—even on single frames. At inference, we propose a depth interpolation method that enables our model to infer high-resolution video depth across sequences of up to 150 frames. Our model outperforms all previous generative depth models in terms of spatial accuracy and temporal consistency. The code and model weights are open-sourced.

# 1 INTRODUCTION

Depth estimation is a foundational problem in understanding the 3D structure of the real world. The ability to accurately perceive and represent depth in video sequences is crucial for a broad range of applications, including autonomous navigation (Borghi et al., 2017), augmented reality (Holynski & Kopf, 2018), and advanced video editing (Zhang et al., 2024; Peng et al., 2024). Although recent advancements in single-image depth estimation (Ke et al., 2024; Yang et al., 2024a; Fu et al., 2024; Ranftl et al., 2021) have led to significant improvements in spatial accuracy, ensuring temporal consistency across video frames remains a substantial challenge.

A major bottleneck in existing video depth estimation (Wang et al., 2023; Shao et al., 2024) is the lack of diverse and large-scale video depth data that capture the complexity of real-world environments. Existing datasets (Geiger et al., 2012; Li et al., 2023; Karaev et al., 2023) are often limited in terms of scale, diversity, and scene variation, making it difficult for models to generalize effectively across different scenarios. From a hardware perspective, depth sensors like LiDAR, structured light systems, and time-of-flight cameras can provide accurate depth measurements but are often costly, limited in range or resolution, and struggle under specific lighting conditions or when dealing with reflective surfaces. Another common approach (Wang et al., 2023; Hu et al., 2024b) is to rely on unlabeled stereo video datasets (Alahari et al., 2013) and state-of-the-art stereo-matching methods (Jing et al., 2024; Xu et al., 2023); however, such methods are complex, computationally intensive, and often fail in areas with weak textures. These limitations hinder the development of robust models that can ensure both spatial precision and temporal consistency in dynamic scenes.

To tackle the challenge, we propose a solution from two complementary perspectives: (1) demonstrating a scalable data collection pipeline to expand data and (2) designing a novel framework that leverages powerful visual priors of generative models to effectively handle various real-world videos.

**Synthetic Data:** Modern virtual environments offer highly realistic graphics and simulate diverse real-world scenarios. For example, driving simulations accurately recreate real-world road conditions, while open-ended virtual worlds depict various complex scenes. Given that modern graphics rendering pipelines often incorporate depth buffers, real-time rendering engines (e.g., Unreal Engine, Unity) offer functionalities for generating and accessing high-quality video depth data, enabling its extraction during interactive sessions. In light of this, we construct DA-V, a synthetic dataset comprising 40,000 video clips collected from diverse interactive virtual environments over two weeks by 10 human players. DA-V captures a wide range of scenarios, covering various lighting conditions, dynamic camera movements, and intricate object interactions in both indoor and outdoor environments, providing the opportunity for models to generalize effectively to real-world environments.

**Framework:** To complement the dataset, we propose a novel framework for video depth estimation that leverages the rich prior knowledge embedded in video generation models. Drawing from recent advancements (Ke et al., 2024), we build upon SVD (Blattmann et al., 2023a) and introduce two key innovations to enhance generalization and efficiency. First, a mixed-duration training strategy is introduced to simulate videos with varying frame rates and lengths by randomly dropping frames. To handle videos of different lengths, those with the same duration are grouped into the same batch, and batch sizes are adjusted accordingly, thus optimizing memory usage and improving training efficiency. Second, a depth interpolation module is proposed, generating intermediate frames conditioned on globally consistent depth estimates from key frames, allowing for high-resolution and coherent inference of long videos under limited computational constraints. Additionally, we refine the pipeline

---

*Equal Contribution. This work was done during his internship at Shanghai AI Laboratory.

by introducing a flow-matching approach (Lipman et al., 2023) and rotary position encoding (Su et al., 2021) to further improve inference efficiency and flexibility.

Our contributions can be summarized as follows:

- We demonstrate a feasible path for scaling depth data from diverse virtual environments and systematically validate that high-fidelity synthetic data can improve the generalization of video depth models to real-world scenarios.
- We propose a new training and inference framework that integrates a mixed-duration training strategy and a long-video inference module, enabling the model to handle varying video lengths while ensuring spatial accuracy and temporal consistency.
- Our method achieves state-of-the-art performance among generative depth models, setting a new benchmark for accuracy and robustness in video depth estimation.

## 2 SYNTHETIC DATA WORKFLOW

**Rendering Pipeline Overview.** Modern rendering pipelines are central to transforming geometric data into visual output, which involves three key stages: 1) Vertex Processing: Geometric data is first processed to determine the position and attributes of each vertex in 3D space. 2) Rasterization: The processed geometric data is then converted into pixels. This stage determines which pixels on the screen correspond to parts of the 3D models, establishing depth and color data for each pixel. 3) Shading: Shaders compute the lighting, coloring, and textures. Deferred shading, a popular technique, separates lighting calculations from geometry rendering to optimize processing. Buffers such as depth buffers store intermediate data essential for accurate lighting and shading.

**Real-time Data Collection.** Following prior works (Richter et al., 2016; Krähenbühl, 2018), we leverage ReShade (ReShade Contributors, 2024) to access depth information available in the rendering process and OBS (Contributors, 2024) to capture synchronized depth and color data from the screen. The collected dataset encompasses diverse scenes and conditions, including modern virtual environments such as *Cities: Skylines II* with its expansive urban landscapes and *Grand Theft Auto V* with its varied open-world settings. Notably, the use and sharing of certain game content are permitted under specific terms and conditions (Rockstar Games, 2024; Paradox Interactive, 2024), including non-commercial use and restrictions on story spoilers. To ensure compliance, we further remove UI elements and on-screen text using a ReShade plugin[1] and manually filter specific scenes to prevent unintended storyline disclosures.

**Data Filtering.** After collecting initial synthetic data, occasional misalignments between the image and depth are observed, especially during menu switches. To filter these frames, we first employ a scene cut method[2] to detect scene transitions based on significant color changes. Then, the depth model (detailed in Sec. 3.1), trained

Table 1: **Comparisons of synthetic datasets**.

| Dataset | Outdoor | Indoor | Dynamic | Video | # Frame |
|---|---|---|---|---|---|
| Hypersim (Roberts et al., 2021) | ✗ | ✓ | ✗ | ✗ | 68K |
| MVS-Synth (Huang et al., 2018) | ✓ | ✗ | ✗ | ✓ | 12K |
| VKITTI (Cabon et al., 2020) | ✓ | ✗ | ✗ | ✓ | 25K |
| MatrixCity (Li et al., 2023) | ✓ | ✗ | ✗ | ✓ | 519K |
| Sintel (Butler et al., 2012) | ✓ | ✗ | ✓ | ✓ | 1.6K |
| DynamicReplica (Karaev et al., 2023) | ✗ | ✓ | ✓ | ✓ | 169K |
| **DA-V (Ours)** | ✓ | ✓ | ✓ | ✓ | **6M** |

on a hand-picked subset of collected data, is used to filter out splited video sequences with low depth metric scores. However, this straightforward approach can lead to excessive filtering of unseen data. We further use a CLIP (Radford et al., 2021) model to compute semantic similarity between the actual and predicted depth, both colorized from a single channel to three. Finally, we uniformly sample 10 frames from each video segment. If both the median semantic and depth metric scores fall below predefined thresholds, the segment is removed. After filtering, we obtain a curated synthetic dataset of 40,000 video clips, each recorded at 30 FPS and lasting 5 seconds, as shown in Table 1.

## 3 GENERATIVE VIDEO DEPTH MODEL

In this section, we introduce Depth Any Video, a generative model designed for robust and consistent video depth estimation. The model builds upon prior video foundation models, framing video depth estimation as a conditional denoising process (Sec.3.1). A mixed-duration training strategy is then

---

[1] https://github.com/4lex4nder/ReshadeEffectShaderToggler
[2] https://github.com/Breakthrough/PySceneDetect

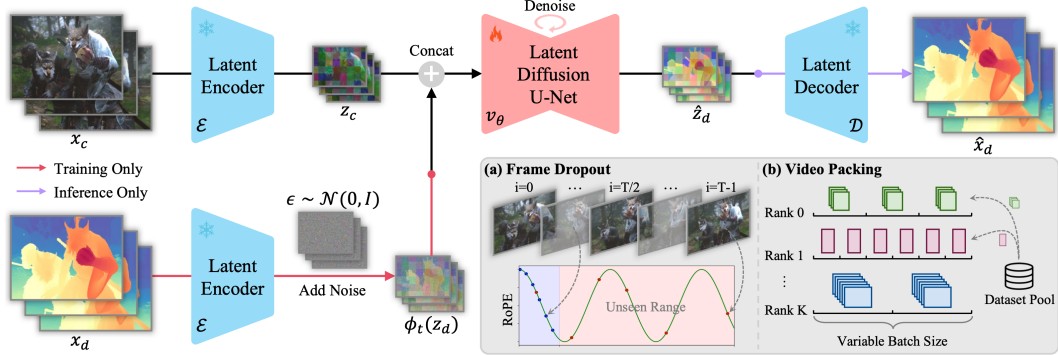

Figure 2: **The overall architecture**. The input video $x_c$ and depth $x_d$ are first encoded into latent space using a pretrained latent encoder $\mathcal{E}$. During training, Gaussian noise $\epsilon$ is added to the latent depth in a forward process, while a denoising model $v_\theta$, conditioned on the latent video $z_c$, removes the noise in a reverse process. After training, the inference flow begins with pure noise, progressively denoises it, and then uses a latent decoder $\mathcal{D}$ to transform it into the prediction depth $\hat{x}_d$ with the original resolution. Besides, the pipeline incorporates a mixed-duration training strategy: **(a) frame dropout** and **(b) video packing** to enhance model generalization and training efficiency.

presented to improve model generalization and training efficiency (Sec.3.2). Finally, we extend the model to estimate high-resolution depth in long videos (Sec. 3.3).

## 3.1 MODEL DESIGN

Our approach builds upon the video foundation model, Stable Video Diffusion (SVD) (Blattmann et al., 2023a), and reformulates monocular video depth estimation as a generative denoising process. The overall framework is illustrated in Figure 2. The training flow consists of a forward process that gradually corrupts the ground truth video depth $x_d$ by adding Gaussian noise $\epsilon \sim \mathcal{N}(0, I)$ and a reverse process that uses a denoising model $v_\theta$, conditioned on the input video $x_c$, to remove the noise. Once $v_\theta$ is trained, the inference flow begins with pure noise $\epsilon$ and progressively denoises it, moving towards a cleaner result with each step.

**Latent Video Condition.** Following prior latent diffusion models (Rombach et al., 2022; Esser et al., 2024), the generation process operates within the latent space of a pre-trained variational autoencoder (VAE), allowing the model to handle high-resolution input without sacrificing computational efficiency. Specifically, given a video depth $x_d$, we first apply a normalization as in Ke et al. (2024) to ensure that depth values fall primarily within the VAE's input range of $[-1, 1]$:

$$\tilde{x}_d = \left( \frac{x_d - d_2}{d_{98} - d_2} - 0.5 \right) \times 2, \tag{1}$$

where $d_2$ and $d_{98}$ represent the 2% and 98% percentiles of $x_d$, respectively. Then, the corresponding latent code is obtained using the encoder $\mathcal{E}$: $z_d = \mathcal{E}(\tilde{x}_d)$. From this latent code, the normalized video depth can then be recovered by the decoder $\mathcal{D}$: $\hat{x}_d = \mathcal{D}(z_d)$. Unlike the recent advanced 3D VAE (Yang et al., 2024c; OpenAI, 2024), which compresses the input across both temporal and spatial dimensions into the latent code, we focus on compressing only the spatial dimension, as in Blattmann et al. (2023a). This is because temporal compression potentially causes motion blur artifacts when decoding latent depth codes, especially in videos with fast motion (detailed in Sec. 4.4).

To condition the denoiser $v_\theta$ on the input video, we first transform the video $x_c$ into latent space as $z_c = \mathcal{E}(x_c)$. Then, $z_c$ is concatenated with the latent depth code $z_d$ frame by frame to form the input for the denoiser. Unlike SVD, we remove the CLIP embedding condition and replace it with a zero embedding, as we find it has minimal impact on performance.

**Conditional Flow Matching.** To accelerate the denoising process, we replace the original EDM framework (Karras et al., 2022) in SVD with conditional flow matching (Lipman et al., 2023), which achieves satisfactory results in just 1 step, compared to the original 25 steps. Concretely, the data corruption in our framework is formulated as a linear interpolation between Gaussian noise $\epsilon \sim \mathcal{N}(0, I)$ and data $x \sim p(x)$ along a straight line:

$$\phi_t(x) = tx + (1 - t)\epsilon, \tag{2}$$

where $\phi_t(x)$ represents the corrupted data, with $t \in [0, 1]$ as the time-dependent interpolation factor. This formulation implies a uniform transformation with constant velocity between data and noise. The corresponding time-dependent velocity field, moving from noise to data, is given by:

$$v_t(x) = x - \epsilon. \tag{3}$$

The velocity field $v_t : [0, 1] \times \mathbb{R}^d \to \mathbb{R}^d$ defines an ordinary differential equation (ODE):

$$d\phi_t(x) = v_t(\phi_t(x))\, dt. \tag{4}$$

By solving this ODE from $t = 0$ to $t = 1$, we can transform noise into a data sample using the approximated velocity field $v_\theta$. During training, the flow matching objective directly predicts the target velocity to generate the desired probability trajectory:

$$\mathcal{L}_\theta = \mathbb{E}_t \left\| v_\theta\left(\phi_t\left(z_d\right), z_c, t\right) - v_t\left(z_d\right) \right\|^2, \tag{5}$$

where $z_d$ and $z_c$ represent the latent depth code and video code, respectively.

## 3.2 MIXED-DURATION TRAINING STRATEGY

Real-world applications often encounter data in various formats, including images and variable-length videos. To enhance the model's generalization across tasks like image and video depth estimation, we implement a mixed-duration training strategy to ensure robustness across various inputs. This strategy includes frame dropout augmentation, which preserves training efficiency when handling long video sequences, and a video packing technique that optimizes memory usage for variable-length videos, enabling our model to scale efficiently across different input formats.

**Frame Dropout.** Directly training long-frame videos is computationally expensive, requiring substantial training time and GPU resources. Inspired by context extension techniques (Chen et al., 2024; 2023; Liu et al., 2024) in large language models, we propose frame dropout augmentation with rotary position encoding (RoPE) (Su et al., 2021) to enhance training efficiency while maintaining adaptability for long videos. Concretely, in each temporal transformer block of the 3D UNet used in SVD, we replace the original sinusoidal absolute position encoding for fixed-frame videos with RoPE to support variable frames. However, training on a short video with RoPE still struggles to generalize to longer ones with unlearned frame positions, as shown in Figure 2(a). To mitigate this, we retain the original frame position indices $i = [0, \cdots, T - 1]$ of the long video with $T$ frames, and randomly sample $K$ frames with their original indices for training. This simple strategy helps the temporal layer generalize effectively across variable frame lengths.

**Video Packing.** To train videos of varying lengths, an intuitive way is to use only one sample per batch, as all data within a batch must maintain a consistent shape. However, this leads to inefficient memory usage for shorter videos. To solve this, we first group videos by similar resolution and crop them to a fixed size. For each batch, we then sample examples from the same group and apply the same frame dropout parameter $K$. The process is illustrated in Figure 2(b). In particular, we increase the batch size for small-resolution and short-duration videos to improve training efficiency.

## 3.3 LONG VIDEO INFERENCE

Using the trained model, we can process up to 32 frames at $960 \times 540$ resolution in one forward pass on a single 80GB A100 GPU. To handle longer high-resolution videos, Wang et al. (2023) applies a sliding window to process short segments independently and concatenate the results. However, this would lead to temporal inconsistencies and flickering artifacts between windows. Thus, we first predict consistent key frames, and then each window generates intermediate frames using a frame interpolation network conditioned on these key frames to align the scale and shift of the depth distributions, as shown in Figure 3. Specifically, the interpolation network is finetuned from the video depth model $v_\theta$ in Sec. 3.1. Instead of conditioning solely on the video, the first and last

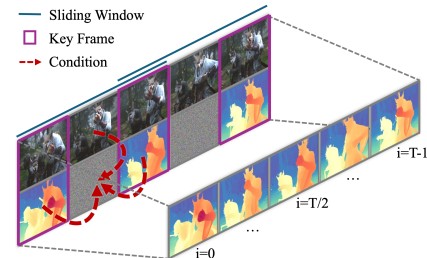

Figure 3: Illustration of the **frame interpolation network**, conditioned on key frames to produce coherent predictions.

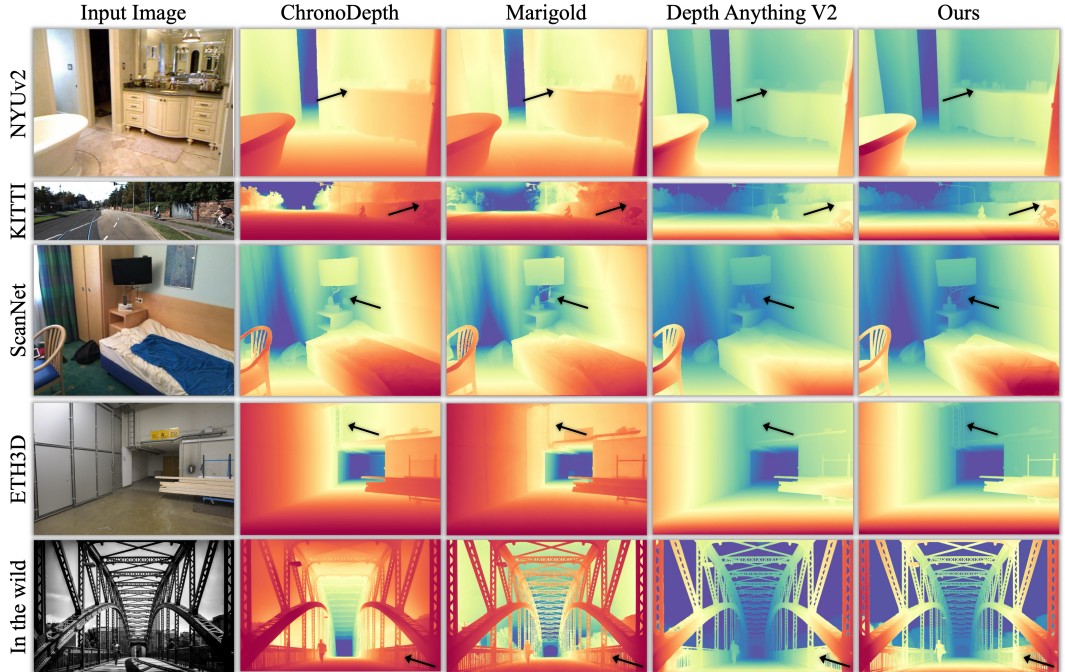

Figure 4: **Qualitative comparisons** of monocular depth estimation methods across different datasets. We are able to capture fine-grained details and generalize effectively on in-the-wild data.

key frames of each window are also used, with a masking map indicating which frames are known. The frame interpolation is formulated as follows:

$$\tilde{z}_d = v_\theta\left(\phi_t\left(z_d\right), z_c, \hat{z}_d, m, t\right), \tag{6}$$

where $\hat{z}_d$ represents the predicted key frames, with non-key frames padded with zeros. The masking map $m$ is used to indicate known key frames, which are set to $1$, while other frames are set to $0$. The masking map is replicated four times to align with the latent feature dimensions. To preserve the pre-trained structure and accommodate the expanded input, we duplicate input channels of $v_\theta$ and halve the input layer's weight tensor as initialization.

## 4 EXPERIMENTS

### 4.1 DATASETS AND EVALUATION METRICS

**Training Datasets.** In addition to the collected DA-V dataset, we follow Ke et al. (2024) by incorporating two single-frame synthetic datasets, Hypersim (Roberts et al., 2021) and Virtual KITTI 2 (Cabon et al., 2020). Hypersim is a photorealistic synthetic dataset featuring 461 indoor scenes, from which we use the official *train* and *val* split, totaling approximately 68K samples. Virtual KITTI 2 is a synthetic urban dataset comprising 5 scenes with variations in weather and camera configurations, contributing about 25K samples to our training.

**Evaluation Datasets.** For monocular depth estimation, we conduct a series of experiments to evaluate our model's performance on four widely used benchmarks. NYUv2 (Silberman et al., 2012) and ScanNet (Yeshwanth et al., 2023) provide RGB-D data from indoor environments captured using Kinect cameras. ETH3D (Schops et al., 2017) features both indoor and outdoor scenes, with depth data collected by a laser scanner. KITTI (Geiger et al., 2012) comprises outdoor driving scenes captured by cameras and LiDAR sensors. For video depth estimation, we sample 98 video clips from ScanNet++ (Yeshwanth et al., 2023), with each clip containing 32 frames. The overlap ratio between adjacent frames in each clip exceeds 40%, ensuring sufficient continuity for video depth estimation.

**Evaluation Metrics.** All evaluations are conducted in the zero-shot setting. Following prior methods (Ke et al., 2024; Fu et al., 2024), we evaluate affine-invariant depth predictions by optimizing for scale and shift between the predicted depth and the ground truth. The quantitative comparisons are

Table 2: **Quantitative comparisons** with state-of-the-art depth estimation methods using single-frame input across four zero-shot affine-invariant depth benchmarks.

| Method | # Training Data | | NYUv2 | | KITTI | | ETH3D | | ScanNet | |
|---|---|---|---|---|---|---|---|---|---|---|
| | Real | Synthetic | AbsRel ↓ | δ1 ↑ | AbsRel ↓ | δ1 ↑ | AbsRel ↓ | δ1 ↑ | AbsRel ↓ | δ1 ↑ |
| *Discriminative Model:* | | | | | | | | | | |
| DiverseDepth (Yin et al., 2020) | 320K | - | 11.7 | 87.5 | 19.0 | 70.4 | 22.8 | 69.4 | 10.9 | 88.2 |
| MiDaS (Lasinger et al., 2019) | 2M | - | 9.5 | 91.5 | 18.3 | 71.1 | 19.0 | 88.4 | 9.9 | 90.7 |
| LeReS (Yin et al., 2021) | 354K | - | 9.0 | 91.6 | 14.9 | 78.4 | 17.1 | 77.7 | 9.1 | 91.7 |
| Omnidata (Eftekhar et al., 2021) | 12.1M | 59K | 7.4 | 94.5 | 14.9 | 83.5 | 16.6 | 77.8 | 7.5 | 93.6 |
| HDN (Zhang et al., 2022) | 300K | - | 6.9 | 94.8 | 11.5 | 86.7 | 12.1 | 83.3 | 8.0 | 93.9 |
| DPT (Ranftl et al., 2021) | 1.4M | - | 9.1 | 91.9 | 11.1 | 88.1 | 11.5 | 92.9 | 8.4 | 93.2 |
| Metric3D (Yin et al., 2023) | 8M | - | 5.8 | 96.3 | 5.8 | 97.0 | 6.6 | 96.0 | 7.4 | 94.1 |
| Depth Anything (Yang et al., 2024a) | 63.5M | - | 4.3 | 98.0 | 8.0 | 94.6 | 6.2 | 98.0 | 4.3 | 98.1 |
| *Generative Model:* | | | | | | | | | | |
| Marigold (Ke et al., 2024) | - | 74K | 5.5 | 96.4 | 9.9 | 91.6 | 6.5 | 96.0 | 6.4 | 95.1 |
| DepthFM (Gui et al., 2024) | - | 63K | 6.5 | 95.6 | 8.3 | 93.4 | - | - | - | - |
| GeoWizard (Fu et al., 2024) | - | 0.3M | 5.2 | 96.6 | 9.7 | 92.1 | 6.4 | 96.1 | 6.1 | 95.3 |
| **Depth Any Video (Ours)** | - | **6M** | **5.1** | **97.0** | **7.3** | **95.1** | **4.7** | **97.9** | **5.3** | **96.6** |

conducted with metrics AbsRel (absolute relative error: $\frac{1}{N}\sum_{k=0}^{N-1}\frac{|\hat{x}_d - x_d|}{x_d}$, where $N$ denoting the number of pixels) and and $\delta 1$ accuracy (percentage of $\frac{1}{N}\sum_{k=0}^{N-1}\max(\frac{\hat{x}_d}{x_d}, \frac{x_d}{\hat{x}_d}) < 1.25$). To assess the temporal consistency of video depth, we further introduce the temporal alignment error (TAE):

$$\text{TAE} = \frac{1}{2(T-2)}\sum_{k=0}^{T-1}\text{AbsRel}\left(f(\hat{x}_d^k, p^k), \hat{x}_d^{k+1}\right) + \text{AbsRel}\left(f(\hat{x}_d^{k+1}, p_-^{k+1}), \hat{x}_d^k\right), \quad (7)$$

where $T$ is the number of frames, $f$ is the projection function that uses transformation matrix $p^k$ to map depth $\hat{x}_d^k$ from the $k$-th frame to the $(k+1)$-th frame, and $p_-^{k+1}$ is the inverse matrix for projection in the reverse direction. The transformation matrix consists of both intrinsic and extrinsic camera parameters, which can be obtained from the dataset.

## 4.2 IMPLEMENTATION DETAILS

Our implementation is based on SVD (Blattmann et al., 2023a), using the diffusers library (von Platen et al., 2022). We employ the AdamW optimizer (Loshchilov & Hutter, 2019) with a learning rate of $6.4 \times 10^{-5}$. The model is trained at various resolutions: $512 \times 512$, $480 \times 640$, $707 \times 707$, $352 \times 1216$, and $1024 \times 1024$, with corresponding batch sizes of 384, 256, 192, 128, and 64. The video length is sampled from 1 to 6, with the batch size adjusting correspondingly to meet GPU memory requirements. Experiments are conducted on 32 NVIDIA A100 GPUs for 20 epochs, with a total training time of approximately 1 day. For training efficiency, we utilize Fully Sharded Data Parallel (FSDP) with ZeRO Stage 2, gradient checkpointing, and mixed-precision training. During inference, we set the number of denoising steps to 3 and the ensemble size to 20 for benchmark comparison, following Ke et al. (2024), to ensure optimal performance. In contrast, the ablation studies do not utilize the ensemble strategy to focus on the isolated impact of individual components. The runtime evaluation is performed on a single NVIDIA A100 GPU with a resolution of $480 \times 640$.

## 4.3 ZERO-SHOT DEPTH ESTIMATION

Our model demonstrates exceptional zero-shot generalization in depth estimation across both indoor and outdoor datasets, as well as single-frame and multi-frame datasets.

**Quantitative Comparisons.** Table 2 presents our model's performance in comparison to state-of-the-art depth estimation models using single-frame inputs. Our model significantly surpasses all previous generative models across various datasets and achieves results that are comparable to, and in some cases better than, those of top-performing discriminative models. For example, compared to

Table 3: **Temporal consistency and spatial accuracy comparisons** on ScanNet++.

| Method | AbsRel ↓ | δ1 ↑ | TAE ↓ |
|---|---|---|---|
| NVDS (Wang et al., 2023) | 22.2 | 61.9 | 3.7 |
| ChronoDepth (Shao et al., 2024) | 10.4 | 90.7 | 2.3 |
| DepthCrafter (Hu et al., 2024b) | 11.5 | 88.1 | 2.2 |
| **Depth Any Video (Ours)** | **9.3** | **93.4** | **2.1** |

Table 4: **Performance and inference efficiency comparisons** on the ScanNet dataset.

| Method | Step ↓ | # Param. ↓ | Runtime ↓ | δ1 ↑ |
|---|---|---|---|---|
| Marigold (Ke et al., 2024) | 50 | 865.9M | 2.06s | 94.5 |
| ChronoDepth (Shao et al., 2024) | 10 | 1524.6M | 1.04s | 93.4 |
| DepthCrafter (Hu et al., 2024b) | 25 | 2156.7M | 4.80s | 93.8 |
| **Depth Any Video (Ours)** | 3 | 1422.8M | **0.37s** | **96.1** |

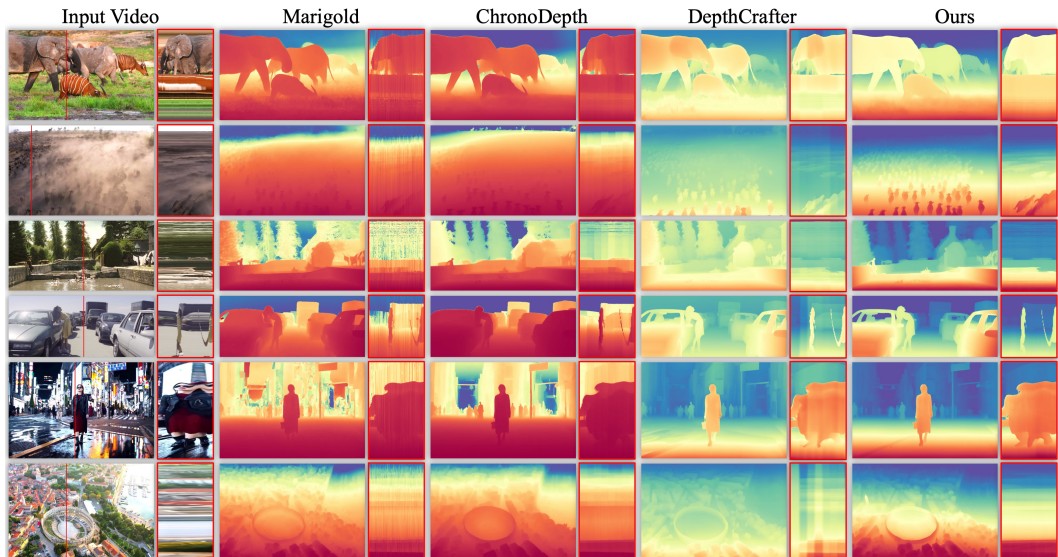

Figure 5: **Qualitative comparisons** of depth estimation models on in-the-wild videos. Red boxes show changes in color or depth over time at vertical red lines in videos. Best viewed by zooming in.

GeoWizard (Fu et al., 2024), our model shows improvements of 0.4 in $\delta 1$ and 0.1 in AbsRel on the NYUv2 dataset, 3.0 in $\delta 1$ and 2.4 in AbsRel on KITTI, 1.8 in $\delta 1$ and 1.7 in AbsRel on ETH3D, and 1.3 in $\delta_1$ and 0.8 in AbsRel on the ScanNet dataset. When compared to Depth Anything (Yang et al., 2024a), we achieve gains of 0.5 in $\delta 1$ and 0.7 in AbsRel on KITTI, along with a 1.5 improvement in the AbsRel metric on the ETH3D dataset. The impressive results are primarily attributed to the large-scale synthetic data we collected. Table 3 presents a comprehensive comparison of our model against previous video depth models. All generative models process multi-frame inputs in a single forward pass. Notably, our model demonstrates improved temporal consistency and spatial accuracy on the ScanNet++ dataset, highlighting its effectiveness in video depth estimation. Table 4 presents detailed comparisons with previous generative methods without ensemble techniques. Our model has fewer parameters than ChronoDepth (Shao et al., 2024) because we utilize a parameter-free RoPE instead of learnable absolute positional embeddings. It also reduces complexity compared to DepthCrafter (Hu et al., 2024b) by removing the clip embedding condition and classifier-free guidance. Additionally, we achieve lower inference time and fewer denoising steps while attaining better spatial accuracy on the ScanNet dataset compared to Marigold, ChronoDepth, and DepthCrafter.

**Qualitative Comparisons.** Figure 4 presents qualitative monocular depth estimation results across different datasets. It highlights the ability of our model to capture fine-grained details compared to Depth Anything V2 (Yang et al., 2024b), such as the cup in the NYUv2 dataset and the ladder in the ETH3D dataset. Moreover, our model handles objects with similar colors more effectively. For example, on the KITTI dataset, the person's head blends with the background, which Depth Anything V2 fails to predict. Compared to generative methods like Marigold and ChronoDepth, our model generalizes well to in-the-wild data, offering a better distinction between the sky and foreground. The DA-V dataset significantly contributes to this, enabling our model to generalize effectively across various environments, particularly in complex, real-world scenarios. Figure 5 further demonstrates qualitative results of depth estimation on open-world videos, covering a wide range of scenarios such as dust and sand, animals, architecture, and human motion presented in both generated and real-world videos. Following Wang et al. (2023), we visualize the changes in estimated

Table 5: **Ablation study** of each component. All variants are trained for 10 epochs to ensure training efficiency. *Memory Util.* refers to the minimum GPU memory utilization across all GPUs, while *Average Metric* represents the average accuracy across four datasets.

| Generative Visual Prior | Conditional Flow Matching | Synthetic Data | Mixed-duration Training | Runtime (s) | Training Time (hours) | Memory Util. (%) | Average Metric AbsRel ↓ | $\delta 1$ ↑ |
|---|---|---|---|---|---|---|---|---|
| ✗ | ✗ | ✗ | ✗ | - | - | - | 21.0 | 65.1 |
| ✓ | ✗ | ✗ | ✗ | 2.4 | - | - | 7.5 | 93.8 |
| ✓ | ✓ | ✗ | ✗ | **0.37** | - | - | 6.9 | 94.9 |
| ✓ | ✓ | ✓ | ✗ | - | 16 | 23 | 6.5 | 95.6 |
| ✓ | ✓ | ✓ | ✓ | - | **12** | **63** | **6.4** | **95.8** |

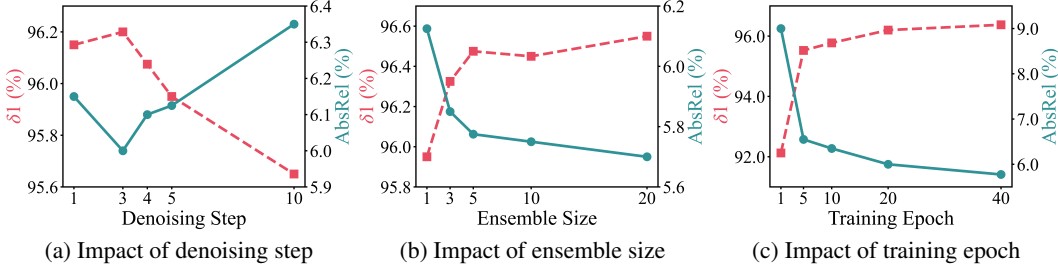

(a) Impact of denoising step · (b) Impact of ensemble size · (c) Impact of training epoch

Figure 6: **Ablation study** of hyper-parameters on depth estimation performance. Average accuracy across four datasets is reported to provide a comprehensive evaluation.

depth values over time at the vertical red lines by slicing along the time axis to better capture temporal consistency. Marigold exhibits zigzag artifacts on a per-frame basis, while ChronoDepth displays similar issues at a per-window level. DepthCrafter, with its large overlap between windows and interpolation of overlap space, achieves smoother transitions across windows but still struggles with window-wise flickering. In contrast, our method ensures global consistency by predicting key frames and interpolating intermediate frames, significantly reducing flicker artifacts between windows.

## 4.4 ABLATION STUDIES

In this section, we evaluate the effectiveness of each component in Depth Any Video. For training efficiency, unless otherwise specified, the model is trained for only 10 epochs during ablation studies.

**Generative Visual Prior.** We investigate the impact of prior visual knowledge from the stable video diffusion model, as shown in Table 5. The first two rows clearly demonstrate that incorporating this prior significantly boosts the model's overall performance. Additionally, Figure 6(c) illustrates that this prior provides a strong initialization, leading to fast training convergence and enabling the model to achieve impressive results with as few as five epochs. This suggests that the generative visual prior not only enhances model performance but also improves training efficiency.

**Conditional Flow Matching.** The second and third rows in Table 5 indicate that flow matching not only reduces inference time, achieving a $6.5\times$ acceleration compared to the original EDM scheduler in SVD, but also results in improvements of 0.6 in AbsRel and 1.1 in $\delta1$. The faster inference is primarily attributed to the ability to achieve strong performance with fewer denoising steps, as shown in Figure 6(a). It demonstrates that even a single denoising step can yield strong results, with the sweet spot identified at three steps for optimal performance. In Figure 6(b), we further evaluate the effectiveness of ensembling multiple predictions, as varying noise initializations produce minor variations in outputs. The results show consistent performance gains as the number of predictions increases, with improvements becoming less pronounced after five predictions.

**Synthetic Data.** In Table 5, the third and fourth rows show that our collected DA-V dataset brings gains of 0.7 in $\delta1$ and 0.4 in AbsRel. The accuracy improvement in outdoor scenes is particularly significant, as shown in Table 2, likely due to the diverse range of outdoor environments in our dataset. Additionally, Figure 4 and 5 demonstrate that models trained on our synthetic data generalize well to in-the-wild scenarios, showcasing both the realism and effectiveness of our synthetic data.

**Mixed-duration Training.** The fourth and fifth entries in Table 5 show that the mixed-duration training strategy improves training efficiency while simultaneously enhancing spatial accuracy. It can save 33% of training time and increase GPU utilization by 40% because different batch sizes can be applied to video sequences of varying lengths, thus optimizing training efficiency. The accuracy

Table 6: **Ablation study** of the reconstruction quality of different variational autoencoders. We categorize the VAEs into 2D and 3D based on the presence of temporal feature interactions.

| Method | Type | AbsRel ↓ | $\delta1$ ↑ |
|---|---|---|---|
| SD2 (Rombach et al., 2022) | 2D | 1.2 | 99.0 |
| SD3 (Esser et al., 2024) | 2D | 0.6 | 99.7 |
| CogVideoX (Yang et al., 2024c) | 3D | 2.2 | 98.6 |
| SVD (Blattmann et al., 2023a) | 3D | 1.5 | 98.1 |

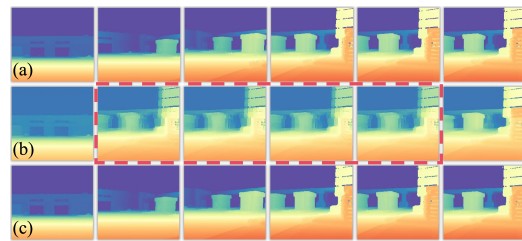

Figure 7: **Visualization** of reconstruction quality.

improvement is attributed to the increased proportion of individual frames during training, achieved by randomly dropping out frames, which enhances the single-frame performance.

**VAE Variants.** Since our depth is generated in the latent space, the quality of the VAE directly impacts the upper bound of the final result. Therefore, we provide a detailed comparison of the depth reconstruction results across different VAEs in Table 6. We find that the 2D VAE surpasses the 3D VAE in reconstruction quality, particularly with the VAE from SD3 (Esser et al., 2024). This indicates there is still potential to further enhance our model's performance. In Figure 7, we present visual comparisons of the reconstructions produced by different 3D VAEs. Specifically, (a) shows the input ground truth depth, (b) displays the reconstruction results from Yang et al. (2024c), which incorporates temporal compression, and (c) shows the results from Blattmann et al. (2023a). Although the VAE with temporal compression can reduce the computational complexity of the latent model, it struggles to handle fast motion effectively, as indicated by the red box in (b).

## 5 RELATED WORK

**Monocular Depth Estimation.** Existing models for monocular depth estimation can be roughly divided into two categories: discriminative and generative. Discriminative models are trained end-to-end to predict depth from images. For example, MiDaS (Lasinger et al., 2019) focuses on relative depth estimation by factoring out scale, enabling robust training on mixed datasets. Depth Anything (Yang et al., 2024a;b) builds on this concept, leveraging both labeled and unlabeled images to further enhance generalization. ZoeDepth (Bhat et al., 2023) and Metric3D (Yin et al., 2023; Hu et al., 2024a) aim to directly estimate metric depth. Generative models (Saxena et al., 2023), such as Marigold (Ke et al., 2024) and GeoWizard (Fu et al., 2024), leverage powerful priors learned from large-scale real-world data, allowing them to generate depth estimates in a zero-shot manner, even on unseen datasets. Our work falls into the second category, but focuses on video depth estimation.

**Video Depth Estimation.** Unlike single-image depth estimation, video depth estimation requires maintaining temporal consistency between frames. To eliminate flickering effects between consecutive frames, some works (Luo et al., 2020; Chen et al., 2019; Zhang et al., 2021) use an optimization procedure to overfit each video during inference. Other approaches (Guizilini et al., 2022; Zhang et al., 2019; Teed & Deng, 2020) directly predict depth sequences from videos. For instance, NVDS (Wang et al., 2023) proposes a refinement network to optimize temporal consistency from off-the-shelf depth predictors. Some concurrent works (Hu et al., 2024b; Shao et al., 2024) have focused on leveraging video diffusion models to produce coherent predictions. However, they often face challenges due to a lack of sufficiently high-quality and realistic depth data.

**Video Generation.** Diffusion models (Ho et al., 2020; Song et al., 2021) have achieved high-fidelity image generation from text descriptions, benefiting from large-scale aligned image-text datasets. Building on this success, VDM (Ho et al., 2022b) first introduces unconditional video generation in pixel space. Imagen Video (Ho et al., 2022a) and Make-a-Video (Singer et al., 2023) are cascade models designed for text-to-video generation. Align Your Latent (Blattmann et al., 2023b) and SVD (Blattmann et al., 2023a) extend Rombach et al. (2022) by modeling videos in the latent space of an autoencoder. Our model builds upon the generative visual prior of SVD, which is trained on diverse real video data, to maintain robust generalization in real-world scenarios.

## 6 CONCLUSION

We present Depth Any Video, a novel approach for versatile image and video depth estimation, powered by generative video diffusion models. Leveraging diverse and high-quality depth data collected from virtual environments, our model could generate temporally consistent depth sequences with fine-grained details across a broad spectrum of unseen scenarios. Equipped with a mixed-duration training strategy and frame interpolation, it generalizes effectively to videos of various lengths and resolutions. Compared to previous generative depth estimation models, our approach sets a new state-of-the-art in performance while significantly enhancing efficiency.

**Limitations.** There are still certain issues in our model, such as difficulties in estimating depth for mirror-like reflections on water surfaces and challenges with extremely long videos. Future work will focus on collecting data for these challenging scenarios and improving model efficiency.

## ACKNOWLEDGEMENTS

This work was supported in part by The National Nature Science Foundation of China (Grant Nos.: 62273301, 62273303), in part by Yongjiang Talent Introduction Programme (Grant No.: 2022A-240-G), in part by Ningbo Key R&D Program (Nos.: 2023Z231, 2023Z229), in part by The National Key R&D Program of China (No. 2022ZD0160101).

## ETHICS STATEMENTS

Our work utilizes game-derived video clips and depth data to develop video depth estimation models. We maintain ethical integrity through three foundational commitments:

1) Legal Data Acquisition: All data was obtained through legally purchased games via authorized platforms (Steam/Epic), with industry-standard capture tools (OBS for video recording). Depth extraction was performed via ReShade's non-invasive post-processing shaders that process decrypted frame buffers provided by the game runtime, with neither cheating methods employed nor reverse engineering of the source code conducted during acquisition.

2) Copyright-Compliant Utilization: Our method complies with research exceptions of international copyright law (such as *the Berne Convention for the Protection of Literary and Artistic Works* and *the WIPO Copyright Treaty*) and aligns with the principles of fair use. Fair use as an exception permitted by law is not unrestrictedly prohibited by the standard clauses in the end user license agreement that have not been negotiated that would be contrary to the goal of the intellectual property legal system in promoting technological innovation and development. Focused exclusively on validating synthetic data pipelines and model training strategies for academic advancement, this non-commercial study does not infringe upon the economic interests or legal rights of right holders.

3) Proactive Responsibility: Although claiming fair use protections, we recognize data ownership resides with copyright holders. We are actively negotiating permissions for potential dataset sharing and will implement strict access control (e.g., institutional authorization) if granted. Any legal consequences arising from this work will be independently borne.

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
