# 1 APPENDIX

## 1.1 EXPERIMENTS

**VAE Variants.** The performance difference between 2D and 3D VAEs primarily stems from two key aspects: training data and network architecture:

1). Image data typically boasts higher aesthetic quality, greater resolution, broader diversity, and larger scale compared to video data, which often has lower resolution, compression artifacts, and significant frame redundancy. Consequently, 2D VAEs generally train more effectively, leveraging these richer image datasets to achieve superior performance.

2). The advanced 2D VAEs trained by SD3 and FLUX increase the number of latent channels to 16. In contrast, the 3D VAEs used in SVD have only 4 channels. As a result, the 2D VAEs can preserve more information in the latent space, thereby significantly enhancing reconstruction performance.

Table 1: **Comparisons** of 2D and 3D VAEs.

| Method | AbsRel ↓ | $\delta 1$ ↑ | TAE ↓ |
|---|---|---|---|
| Single-frame Decoder | 9.5 | 93.1 | 2.3 |
| **Multi-frame Decoder** | **9.3** | **93.4** | **2.1** |

To further examine the ability of 3D VAEs to capture 3D information, we design an experiment detailed in Table 1. Specifically, this involves decoding sequence latents frame-by-frame (as depicted in the first row) and decoding all latent frames in a single pass (as shown in the second row). The results demonstrate that incorporating temporal information is beneficial for enhancing the reconstruction capability.

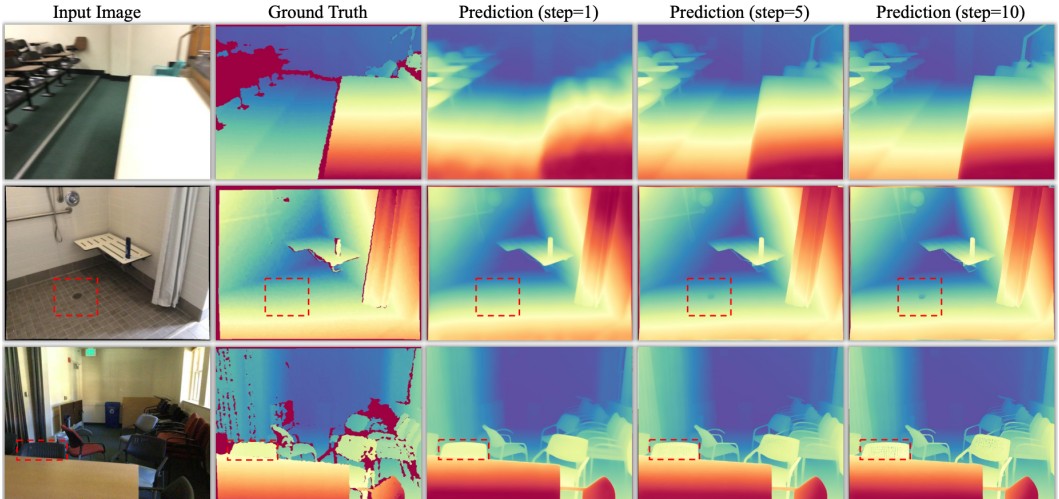

Figure 1: **Qualitative comparisons** of different denoising steps.

**Conditional Flow Matching.** We present a qualitative comparison of denoising steps in Figure 1. Increasing the number of denoising steps can improve the geometric quality, resulting in sharper and more detailed geometry, as shown in the first row. However, fine-grained geometry does not always result in a better depth metric due to the discrepancy between the synthetic data used for training and the real-world data used for evaluation. For example, the red boxes in the second and third rows highlight the model's ability to predict the perforated structure of the drain and the chair's backrest. In contrast, the ground truth captured by the depth sensor applies filtering or interpolation to smooth this region, creating a gap between the real-world data and the synthetic data.

Table 2: **Ablation study** of denoising steps.

| Denoising Step | 1 | 2 | 3 | 4 | 5 | 10 | 20 | 50 |
|---|---|---|---|---|---|---|---|---|
| AbsRel ↓ | 3.74 | 3.80 | 3.06 | 2.62 | 2.39 | 1.92 | 1.88 | 1.84 |

We further evaluate the effect of denoising steps on the synthetic dataset, i.e., Hypersim, as shown in Table 2. We find that increasing the number of denoising steps can improve accuracy; however, the improvement saturates after reaching a certain step.

## 1.2 Implementation Details

For the video interpolation model, the key frames are concatenated with noise along the latent channels, serving as the conditioning input. The mask map is replicated four times to match the latent channel dimensions, simplifying the implementation when initializing the input convolution weights.