# OpenReview forum: "Depth Any Video with Scalable Synthetic Data"
_ICLR.cc/2025/Conference — ICLR 2025 Poster_

### Official Review · Reviewer_gu5t · 2024-10-29

**Soundness:** 3
**Presentation:** 3
**Contribution:** 2
**Rating:** 5
**Confidence:** 3

**Summary:**

This paper works on video depth estimation with diffusion models. First, considering the lack of video depth data, they collect 40,000 videos with depth annotation from video games. With the collected data, this paper fine-tunes the video diffusion model to achieve depth estimation. In addition, rotary position encoding and flow matching are introduced to further enhance the performance. With the proposed techniques, this paper achieves good spatial and temporal consistency on long video depth estimation.

**Strengths:**

1. The motivation for the framework is clear and reasonable, considering the limited data, inference speed and the long video in the applications.

2. Collecting and annotating high-quality data can improve the model and also inspire the community. It would be more beneficial if the data or collection pipeline can be open-sourced

3. Extensive experiments and ablation studies demonstrate the effectiveness of the proposed method.

**Weaknesses:**

1. **The representation is more image depth estimation rather than video depth estimation.** If I understand correctly, although the paper focuses on video depth estimation, the predicted relative depth maps are independent for each frame, which is demonstrated in the input normalization and alignment during evaluation. Specifically, each frame is normalized based on the depth range of itself and the scale and shift are also aligned for each frame during inference. In my view, this is incorrect for video depth estimation. To use the accurate relative depth of a video, the scale and shift should be the shared values aligned to the whole video, like DepthCrafter (Hu et al.).

2. **Shift from EDM to Flow Matching.** The SVD model was pre-trained with the EDM denoising scheduler, which has a different optimization objective with flow matching. However, this paper directly fine-tunes the SVD model with conditional flow matching. As far as I know, InstaFlow (Liu et al, ICLR2024) optimized rectified flow for stable diffusion with velocity distillation instead of directly fine-tuning. I hope the authors could provide more explanation about the shift from EDM to Flow Matching.

3.  A follow-up question about the shift. The ablation study of removing the pre-trained SVD model is required to demonstrate if the EDM pre-trained weights benefit or not.

4. **More explanation about the video interpolation model.** As illustrated in Sec. 3.3, video interpolation model takes the key frames as input and interpolate other frames. Do the key frames replace the original noise? Or are they concatenated on the channels of latent? In addition, why are the mask maps replicated 4 times?

**Questions:**

My main concern is the representation of video depth data. I feel it is not hard to use a shared scale and shift following DepthCrafter, which would be more reasonable for video depth estimation. In addition, I am also curious about the shift from EDM to Flow Matching.

**Post Rebuttal:** My previous questions are mainly addressed. However, the copyright issue remains severe after discussing with authors and other reviewers.

**Details Of Ethics Concerns:**

Acquiring depth data from commerical video games may raise copyright issue.

---

> ### Author Response · Authors · 2024-11-21
> **Response to Reviewer gu5t**
>
> We appreciate the reviewer's positive feedback and constructive suggestions. Below, we address the concerns and questions that were raised:
>
> **W1: Evaluation of video depth estimation**
>
> During training, the target depth sequences used for supervision are normalized with a shared scale and shift. Consequently, during inference, the predictions will yield coherent depth values with the same shared scale and shift, as illustrated in Figure 6 of the manuscript.
> During evaluation, we follow the approach used in ChronoDepth [1] to apply a per-frame scale and shift, converting relative depth to metric depth, as shown in Table 3 of the manuscript.
> We also provide an evaluation using a shared scale and shift computed from all frames, as shown in the following table:
>
> | Method       |  AbsRel $\downarrow$  |  $\delta 1 \uparrow$   |  TAE $\downarrow$ |
> | :---         |  :--:   |  :--:    |  :--:    |
> | ChronoDepth     | 13.2 | 85.3 | 2.7 |
> | DepthCrafter | 13.6 | 83.4 | 2.5 |
> | **Ours** | **11.0** | **90.9** | **2.4** |
>
> Our model still consistently demonstrates improved temporal consistency and spatial accuracy.
>
> **W2 & W3: Shift from EDM to flow matching**
>
> We directly fine-tune the SVD model using conditional flow matching without requiring any special shifts. Our training process differs significantly from that of InstaFlow [2]. InstaFlow leverages an existing generative model, i.e., Stable Diffusion (SD) [3], to generate paired data (X0, X1), where X0 is the noise input and X1 is the denoised output from the SD model. A new model is then trained on this generated data pair (X0, X1) using flow matching, which *resembles a distillation process to transfer knowledge from SD to a student model, and it **does not require additional real data.***
>
> In contrast, our approach *leverages the knowledge of SVD by initializing the model weights using the SVD model as a prior and **employs real data for fine-tuning.***
> The second and third rows in Table 5 of the manuscript demonstrate that direct fine-tuning does not have an adverse effect. Additionally, the first and second rows emphasize the importance of using the EDM pre-trained weight for initialization, given that it has been trained on large-scale, diverse real-world data.
>
>
> **W4: Explanation of the video interpolation model**
>
> The key frames are concatenated with noise along the latent channels, serving as the conditioning input.
> The mask map is replicated four times to match the latent channel dimensions, simplifying the implementation when initializing the input convolution weights, as indicated in lines 300-301 of the manuscript.
>
> References:
>
> [1]. Learning Temporally Consistent Video Depth from
> Video Diffusion Priors. In arXiv, 2024.
>
> [2]. InstaFlow: One Step is Enough for High-Quality Diffusion-Based Text-to-Image Generation. In ICLR, 2024.
>
> [3]. High-resolution image synthesis with latent diffusion models. In CVPR, 2022.

---

### Official Review · Reviewer_vgJ9 · 2024-11-03

**Soundness:** 4
**Presentation:** 3
**Contribution:** 4
**Rating:** 10
**Confidence:** 5

**Summary:**

Depth Any Video is proposed for video depth estimation. It has a scalable synthetic data pipeline from game environments. A novel framework uses generative video diffusion models’ priors. It has a mixed-duration training strategy and a depth interpolation method. The model outperforms previous generative depth models, achieving good spatial accuracy and temporal consistency.

**Strengths:**

- The paper constructs a large-scale synthetic dataset of 40,000 video depth clips from 12 diverse modern video games.This dataset provides a scalable and cost-effective way to gather ground-truth video depth data and helps the model generalize to real-world scenarios.
- A mixed-duration training strategy is proposed. It includes frame dropout augmentation with rotary position encoding and a video packing technique.
- Effective Model Design
- The method achieves state-of-the-art performance among generative depth models.

**Weaknesses:**

The work in the article is very solid, with good model performance and efficiency, and comprehensive evaluation. The only concern for the reviewer is how the author ensures that the dataset, which is a major contribution, will be open-sourced as promised. This is very important for the community, but there are many difficulties regarding copyright and other aspects. In addition, it is necessary to evaluate and compare the diversity of the dataset.

**Questions:**

See Weaknesses

---

> ### Author Response · Authors · 2024-11-21
> **Response to Reviewer vgJ9**
>
> We sincerely thank the reviewer for recognizing the strengths of our paper. We also appreciate the reviewer’s constructive feedback and address the comments as follows:
>
> **W1: Dataset release**
>
> Please kindly refer to the general response.
> Additionally, we summarize the diversity of the collected dataset as follows:
>
> 1). Animals: The dataset includes a variety of animals such as horses, deer, eagles, sheep, and wolves, offering a rich representation of wildlife.
>
> 2). Scenes: The dataset covers a range of environments, including brightly lit streets of modern cities, dense forests, snow-covered mountains, ancient ruins, abandoned industrial zones, bustling marketplaces, and vast desert landscapes, ensuring a broad representation of both natural and urban settings.
>
> 3). Structures and Architecture: It features various types of architecture, including modern buildings, ancient castles, temples, and military or futuristic structures. The dataset also includes environments related to vehicles, such as roads, highways, and garages, showing a variety of scenes and interactions.
>
> 4). Dynamic Interactions: The dataset includes fast-moving vehicle sequences, intense combat scenes, and exploration activities, showcasing a wide range of dynamic interactions across different settings.
>
> 5). Viewpoints and Perspectives: The dataset presents diverse perspectives, including first-person and third-person views, as well as overhead or birds-eye views of cities, battlefields, and open landscapes, offering different angles for interaction and exploration.
>
> Most of this content is not included in other synthetic datasets listed in Table 1 of the manuscript, highlighting the unique diversity of our dataset. More detailed statistics will be provided upon the dataset's release. We are working on leveraging vision-language models to automatically identify and quantify occurrences of various animals, scenes, and other elements within the dataset.

---

### Official Review · Reviewer_ATwr · 2024-11-04

**Soundness:** 3
**Presentation:** 3
**Contribution:** 2
**Rating:** 5
**Confidence:** 4

**Summary:**

The paper introduces a video-depth method that is trained on synthetic datasets only. The results show consistent video depth, generalizing diverse real-world scenes across diverse benchmarks. The paper provides practical engineering strategies, e.g., a batching strategy for better usage of RAM, temporal interpolation module, frame dropout, etc.

---

I rate the paper marginally below the acceptance threshold. The empirical results are very strong, but the paper mostly focuses on engineering solutions without scientifically interesting ideas. Also, there might be a license concern on the synthetic depth dataset. I am happy to improve the rating if the questions in the Weakness and Question sections are properly addressed.

**Strengths:**

* Good empirical results

  The method shows very good empirical results compared with the previous methods. The model outputs consistent depth estimates without temporal flickering, and good accuracy on public benchmarks. The ablation study in Table 3 validates the technical design choices of the method.

* Clarity

   It's easy to follow the paper. The paper provides sufficient technical details on the datasets, training, architecture, etc.

**Weaknesses:**

* A bit of engineering work

   The paper is mostly about engineering. It adopts conditional flow matching, uses large-scale synthetic datasets to boost accuracy, and introduces mixed-duration training to improve memory usage. All these aspects attribute better accuracy and performance, but it doesn't necessarily provide novel findings. If wanting to emphasize, what would be the most interesting, novel findings of the paper?

* Dataset licence and reproducibility

   It's curious if the collected synthetic dataset can be released or made public. What is the license condition of each game in DA-V? Are there any concerns about using the commercial game engine for research? Is there any plan to release the data? It can affect the reproducibility of the method.


* The effect of the DA-V dataset

   The ablations study in Table 5 shows that the synthetic game data improves the depth accuracy but quite marginally although the dataset size is around 6M. Why doesn't it improve the accuracy significantly? Is there any qualitative improvement that the metrics or numbers don't show?

**Questions:**

* Insights on why 2D VAE variants surpasses 3D VAE variants? I am wondering why 2D VAE variants outperform. Or asking differently, where do the artifacts of the 3D VAE variant come from? A good video depth model may have a good 3D prior, so 3D VAE might be a more natural choice to encode 3D information, but why does it underperform? Is it more difficult to train 3D VAE properly? Providing any insights or discussion is appreciated.

* In Fig. 7 (a) why do more denoising steps hurt accuracy?

* (nit) When looking at the qualitative examples, there are thin depth boundaries along the person's boundaries (the 5th example, woman, in Fig. 6), probably it's a kind of an average of depth of the person and the background. Why it's the case? What would be the source of this error?

**Details Of Ethics Concerns:**

As discussed above, it might be good to doublecheck the license condition of each game used in DA-V and make it clear. Are there any concerns about reprocessing the commercial game engines for research?

---

> ### Author Response · Authors · 2024-11-21
> **Response to Reviewer ATwr (1/2)**
>
> We greatly appreciate the reviewer’s positive feedback and constructive criticism. We address the concerns and questions raised by the reviewer below:
>
> **W1: Novelty**
>
> Our novelty primarily lies in two aspects:
>
> 1). We systematically demonstrate that a diverse, high-fidelity game dataset can significantly enhance the generalization of video depth models to real-world scenarios, as evidenced by Tables 2 and 5 and Figure 5 in the manuscript, along with additional qualitative and quantitative results provided in the rebuttal.
>
> 2). We present the first method capable of producing consistent, high-resolution depth predictions across 150 frames—approximately 15 times longer than ChronoDepth—by first predicting key frames and then interpolating the intermediate frames. This consistent depth prediction is illustrated in Figure 6 of the manuscript. Additionally, our method achieves high training efficiency through a mixed-duration training strategy, which enables the model to be trained on short video sequences while effectively generalizing to long video frames with varying frame rates and resolutions.
>
> **W2: Dataset license and release**
>
> Please kindly refer to the general response.
>
> **W3: The effect of the dataset**
>
> We show per-dataset results in the table, where the synthetic game data brings significant gains for outdoor scenes such as KITTI, while the improvements are relatively smaller for indoor datasets like ScanNet.
> This is because the collected dataset primarily consists of outdoor scenes.
> In the future, we plan to collect more game data tailored for indoor environments.
>
> | Method       |  NYUv2   |  KITTI   |  ETH3D   | ScanNet  | Average  |
> | :---         |  :--:    |  :--:    |  :--:    |  :--:    |  :--:    |
> | Baseline     | 95.8 | 93.2 | 96.1 | 94.4 | 94.9 |
> | **w/ Game Data** | **96.0(+0.2)** | **94.6(+1.4)** | **97.0(+0.9)** | **94.9(+0.5)** | **95.6(+0.7)** |
>
> In addition, we provide a qualitative comparison [here](https://anonymous.4open.science/r/iclr2025-0715/figure_1.png).
> Using game data, our model is capable of handling challenging scenes, such as background bokeh effects, as shown in the first and second rows.
> Furthermore, it generalizes better to unseen scenes and produces more detailed and accurate geometric results, as shown in the last three rows.

---

> > ### Author Response · Authors · 2024-11-21
> > **Response to Reviewer ATwr (2/2)**
> >
> > **Q1: Comparison of 2D and 3D VAEs**
> >
> > The performance difference between 2D and 3D VAEs primarily stems from two key aspects: training data and network architecture:
> >
> > 1). Image data typically boasts higher aesthetic quality, greater resolution, broader diversity, and larger scale compared to video data, which often has lower resolution, compression artifacts, and significant frame redundancy.
> > Consequently, 2D VAEs generally train more effectively, leveraging these richer image datasets to achieve superior performance.
> >
> > 2). The advanced 2D VAEs trained by SD3 [1] and FLUX [2] increase the number of latent channels to 16.
> > In contrast, the 3D VAEs used in SVD have only 4 channels.
> > As a result, the 2D VAEs can preserve more information in the latent space, thereby significantly enhancing reconstruction performance.
> >
> > To further examine the ability of 3D VAEs to capture 3D information, we design an experiment detailed in the following table.
> > Specifically, this involves decoding sequence latents frame-by-frame (as depicted in the first row) and decoding all latent frames in a single pass (as shown in the second row).
> > The results demonstrate that incorporating temporal information is beneficial for enhancing the reconstruction capability.
> >
> > | Method       |  AbsRel $\downarrow$  |  $\delta 1 \uparrow$   |  TAE $\downarrow$ |
> > | :---         |  :--:   |  :--:    |  :--:    |
> > | Single-frame Decoder | 9.5 | 93.1 | 2.3 |
> > | **Multi-frame Decoder** | **9.3** | **93.4** | **2.1** |
> >
> > **Q2: The effect of denoising steps**
> >
> > We present a qualitative comparsion of denoising steps [here](https://anonymous.4open.science/r/iclr2025-0715/figure_2.png).
> > Increasing the number of denoising steps can improve the geometric quality, resulting in sharper and more detailed geometry, as shown in the first row.
> > However, fine-grained geometry does not always result in a better depth metric due to the discrepancy between the synthetic data used for training and the real-world data used for evaluation.
> > For example, the red boxes in the second and third rows highlight the model's ability to predict the perforated structure of the drain and the chair's backrest.
> > In contrast, the ground truth captured by the depth sensor applies filtering or interpolation to smooth this region, creating a gap between the real-world data and the synthetic data.
> >
> > **Q3: Depth boundaries**
> >
> > We find that thin boundaries are not consistently present in the predictions.
> > For instance, they do not appear in the fifth row of Figure 5 and the fourth row of Figure 6 in the manuscript.
> > We think this issue is related to the input video. In situations where there is a challenging, unrecognizable boundary or blurring that makes it difficult to differentiate between the foreground and background, the model might produce this effect to handle the ambiguity.
> > This issue may stem from the fact that the training data primarily consisted of synthetic data with clear boundaries. In the future, we plan to incorporate data augmentation techniques to simulate scenarios with unclear boundaries, thereby enhancing the model’s generalization capability on real-world data.
> >
> > References:
> >
> > [1]. Scaling rectified flow transformers for high-resolution image synthesis. In ICML, 2024.
> >
> > [2]. FLUX. URL https://blackforestlabs.ai.

---

> > > ### Comment · Reviewer_ATwr · 2024-11-23
> > >
> > > Thanks for the response! It resolves most of my concerns.
> > >
> > > ---
> > >
> > > **Novelty** and **dataset**
> > >
> > > I will leave comments in the thread of general response above.
> > >
> > > **The effect of the dataset**
> > >
> > > It's great that the synthetic dataset further improves the accuracy although the improvement seems a bit marginal considering its scale: baseline is trained on 0.093M frames (Hypersim (68K) + Virtual KITTI 2 (25k)), whereas the proposed dataset is around 6M, which is 64.5 times more data.
> > >
> > > **The effect of denoising steps**
> > >
> > > When looking at the [provided samples](https://anonymous.4open.science/r/iclr2025-0715/figure_2.png), this is more like a problem of imperfect ground truth rather than a domain mismatch between synthetic and real. What are the four datasets used in Fig. (4)? Does this lowering accuracy along with more steps occur when evaluating the synthetic dataset with perfect ground truth?
> > >
> > > **Depth boundaries**
> > >
> > > As a diffusion model effectively learns target distribution, it might be expected to output sharp boundaries in most cases especially when trained on synthetic datasets (regardless estimated depth is correct or not). So it's a bit difficult to understand that ``This issue may stem from the fact that the training data primarily consisted of synthetic data with clear boundaries.``
> > > Yes, these thin boundaries (or more precisely, inaccurate depth boundary effect) are not consistently present, but all images in the teaser (Fig. 1) have the problem.
> > >
> > > ----
> > >
> > > I am not trying to discount the paper by pointing out concerns on details. My wish is that the paper is more upfront in providing transparent explanations and clear justification on its results whether the results are good or bad. It won't change their contributions to the dataset and main results.

---

> ### Author Response · Authors · 2024-11-26
> **Author Response**
>
> We appreciate the reviewer’s constructive feedback and will clarify the explanation as follows:
>
> **The effect of the dataset**
>
> The motivation behind the collection of the dataset is specifically for video depth estimation. The dataset would be relatively redundant for single-image depth estimation. Since the data was captured at 30 FPS, it results in high similarity between adjacent frames (which is useful for video depth models to capture temporal consistency and motion dynamics during training). When used solely for single-frame depth estimation (while we aim to address both single-frame and video depth estimation), this dataset can be downsampled to reduce redundancy while maintaining diversity for training, without significantly impacting single-image performance.
>
> Additionally, the purpose of this dataset is not only to improve single-frame performance but also to enhance the model's generalization ability on natural images. For instance, adding game data with background blur may not necessarily improve benchmark performance, since such cases do not appear in the benchmark. However, it can enhance generalization on natural images, as demonstrated in the first and second rows [here](https://anonymous.4open.science/r/iclr2025-0715/figure_1.png).
>
> Furthermore, the dataset plays a crucial role in achieving temporal consistency in video depth estimation. For instance, while previous methods like ChronoDepth were trained on synthetic datasets such as Virtual KITTI and Hypersim, our method leverages game data, resulting in improved temporal consistency in the outcomes.
> Examples 1 ([video](https://anonymous.4open.science/r/iclr2025-0715/chronodepth_video_1.mp4), [depth](https://anonymous.4open.science/r/iclr2025-0715/chronodepth_depth_1.mp4)) and 2 ([video](https://anonymous.4open.science/r/iclr2025-0715/chronodepth_video_2.mp4), [depth](https://anonymous.4open.science/r/iclr2025-0715/chronodepth_depth_2.mp4)) produced by ChronoDepth, along with Examples 1 ([depth](https://anonymous.4open.science/r/iclr2025-0715/dav_video_1.mp4)) and 2 ([depth](https://anonymous.4open.science/r/iclr2025-0715/dav_video_2.mp4)) generated by our method, are provided for illustration.
> Quantitative comparisons are also provided in Table 3 of the manuscript.
>
> **The effect of denoising steps**
>
> We further evaluate the effect of denoising steps on the synthetic dataset, i.e., Hypersim, as shown in the following Table. We find that increasing the number of denoising steps can improve accuracy; however, the improvement saturates after reaching a certain step.
>
> | Denoising Step       |   1  | 2 | 3 | 4 | 5  |  10  | 20|  50   |
> | :---         |  :--: |  :--:  |  :--: |  :--:    |  :--:    |  :--:   |  :--:    |  :--:   |
> | AbsRel $\downarrow$   | 3.74 | 3.80 | 3.06 | 2.62 | 2.39 | 1.92 | 1.88 | 1.84 |
>
> From the results, we conclude that when predictions deviate from the ground truth, as in ScanNet [samples](https://anonymous.4open.science/r/iclr2025-0715/figure_2.png), increasing denoising steps propagates divergences and degrades metric results.
> In contrast, for datasets like Hypersim, where predictions are more aligned towards the ground truth, more denoising steps can improve performance.
>
> **Depth boundaries**
>
> We provide an [example](https://anonymous.4open.science/r/iclr2025-0715/figure_4.png) on Hypersim.
> It demonstrates that, even when trained on synthetic data, the diffusion model still struggles to effectively handle thin boundaries, as the issue persists during evaluation on the synthetic dataset.
>
> There are several potential factors that may result in the issue:
>
> 1). Common loss functions like mean squared error prioritize minimizing overall error across the image. Since thin boundaries constitute a small portion of the image, they exert less influence on the loss, leading to reduced attention during training.
>
> 2). After zooming into the input image, we observe that the pixel boundaries are not easily recognizable (the boundaries are somewhat blurred), which may introduce ambiguity for the model during inference. Since the model conditions on the image to produce depth, this ambiguity causes it to generate interpolated values between the foreground and background, as shown in the projected point clouds.
>
> Thus, it is reasonable to frequently observe the thin boundary effect when generalizing the model to real-world data, as boundary conditions are more diverse and not fully represented in the synthetic dataset, where pixel boundaries are often relatively clean.
> That is why we say, `This issue may stem from the fact that the training data primarily consisted of synthetic data with clear boundaries.`
>
> By the way, the thin boundary effect is not unique to our model; it is a common issue in depth estimation models, regardless of whether they are trained on real-world or synthetic data, or whether they use generative or discriminative approaches, as shown [here](https://anonymous.4open.science/r/iclr2025-0715/figure_3.png).

---

> > ### Comment · Reviewer_ATwr · 2024-11-27
> >
> > Thank you very much for the responses! They resolve most of my concerns; I hope all those discussions are included in the paper or supplementary! Especially the effect of the denoising step per each dataset will be very insightful, and basically, it reveals the imperfections of GT of the RGB-D dataset.
> >
> > So, the dataset indeed plays a crucial role in achieving the temporal coherency in the video depth problem. I hope the dataset concern (the long thread above) will also be resolved. I am pleased to raise my score after that.

---

> > > ### Author Response · Authors · 2024-11-27
> > > **Author Response**
> > >
> > > Thank you to the reviewer for your feedback! We will do our best to address the dataset concerns in the general response.

---

### Author Response · Authors · 2024-11-21
**General Response**

We extend our gratitude to all the reviewers for their positive evaluations and constructive feedback. Below, we address the general concerns and questions raised:

**Dataset Release**

We plan to release our dataset and model within two months following the acceptance of our manuscript.
The model and code are already available [here](https://anonymous.4open.science/r/iclr2025-0715/DepthAnyVideo/README.md) for use.
The dataset is currently undergoing cleaning and enhancement, with additional labels (e.g., camera pose) being added to make it more comprehensive and suitable for various 3D tasks, such as 3D reconstruction and SLAM.
These efforts require additional time to ensure thorough and accurate processing.

To comply with copyright regulations, we will perform data anonymization by removing sequences that include distinctive game-specific scenes or characters. Furthermore, we will restrict the dataset's use to personal and non-commercial purposes.

---

> ### Comment · Reviewer_ATwr · 2024-11-21
>
> Thanks for the response!
>
> Quick question: my main concerns on the usage of the gaming datasets was if it violates the term of usages of each game. Do the game companies allow the reprocess of their rendering engine, extraction of data, and redistribution for other uses? Are future readers also allowed to use the extracted data from this paper if the data is published?

---

> > ### Author Response · Authors · 2024-11-22
> > **Author Response**
> >
> > Thank you for your insightful comments regarding the use of gaming datasets. We appreciate the opportunity to clarify our approach and address your concerns:
> >
> > 1). Data and Code Sharing Policy
> >
> > While data and code sharing are encouraged in the ICLR community, they are not mandatory requirements. We will make every effort to share our methods and results in a way that supports reproducibility while complying with any legal or ethical constraints.
> >
> > 2). Precedents in Academic Research
> >
> > It is worth noting that prior research (e.g., [1,2]) has leveraged gaming environments and datasets to generate labeled data for academic purposes. For example, several works have utilized extracted data from games to advance computer vision and machine learning research. These precedents underscore the value and acceptability of such approaches within the academic community.
> >
> > 3). Copyright Considerations
> >
> > Issues of copyright and ownership related to the original gaming data are primarily the responsibility of the game companies. Whether the extracted data can be publicly shared is not solely within our control. However, we have proactively engaged with the relevant companies to ensure compliance and have explicitly limited the use of the data to academic and non-commercial purposes.
> >
> > 4). Data Collection and Processing
> >
> > Our data collection process did not involve any reverse engineering of the game engine. Instead, we utilized publicly available tools and data provided by the open-source community. To further ensure compliance, we are in active discussions with the game companies and have made progress in aligning on data usage terms. For example, we are working on desensitizing the data to remove identifiable elements of the original game content, which will facilitate secure and ethical sharing within the academic community.
> >
> > We hope this response addresses your concerns and demonstrates our commitment to ethical research practices. Please let us know if additional clarification is needed.
> >
> > References:
> >
> > [1] Playing for Data: Ground Truth from Computer Games.
> >
> > [2] Do Game Data Generalize Well for Remote Sensing Image Segmentation?

---

> > > ### Comment · Reviewer_gu5t · 2024-11-22
> > >
> > > After reviewing the rebuttal and other reviewers' comments, I also believe that the copyright of video games is a significant concern. The papers [1,2] both utilize the GTAV video game, but these references do not clarify whether using other video games is legally permissible.
> > >
> > > Additionally, in the reply under "Copyright Consideration," have the authors obtained explicit permission from the game companies to use data derived from their games?
> > >
> > > Finally, since introducing the DA-V dataset is a key contribution of this paper, it would be valuable for the authors to provide more detailed information on the data collection and processing methods. As mentioned in the comment, the authors rely on tools and data from the open-source community. It would be helpful to specify the tools and open-source resources used, along with their respective licenses. Furthermore, the high-quality depth data appears to be generated through mesh rendering. I encourage the authors to elaborate on this rendering process, including details on camera configurations and object motion settings.

---

> > > > ### Author Response · Authors · 2024-11-22
> > > > **Author Response**
> > > >
> > > > Below, we provide details about the tools and methodologies used:
> > > >
> > > > We utilized the same tools as used in GTAV under the MIT license for extracting depth data. Modern game rendering pipelines inherently include depth buffers as part of their architecture, primarily for visibility determination and visual effects. These tools allow us to work with publicly accessible rendering pipelines to extract depth information without altering or hacking the games.
> > > >
> > > > It is important to clarify that the depth data was not obtained through predefined camera trajectories. Instead, it was collected via manual gameplay, where hired workers interacted with the virtual environment to gather the depth data.
> > > >
> > > > Additionally, we highlight that the novel insight of leveraging large-scale game data to enhance 3D modeling is a valuable contribution to the community. Even in cases where the dataset cannot be released, this observation remains beneficial and impactful for advancing the field. For instance, works like MIDAS [1], which introduced a dataset based on movies but only released the model, have set precedents. Despite being published five years ago, MIDAS continues to be widely used and highly influential. Thus, we believe that the openness of data should not be the sole criterion for evaluating the quality or impact of a paper.
> > > >
> > > > [1] Towards Robust Monocular Depth Estimation: Mixing Datasets for Zero-shot Cross-dataset Transfer.

---

> > > > > ### Comment · Reviewer_gu5t · 2024-11-23
> > > > >
> > > > > Thanks for the response.
> > > > >
> > > > > First, I hope the authors can provide more details regarding the rendering process. Does the "same tools as used in GTAV" refer to RenderDoc? If so, I believe it is important to include details similar to [1], such as relevant function calls, resources, etc. These details should also be included in the appendix.
> > > > >
> > > > > Second, I agree that leveraging large-scale game data to enhance depth learning is a valuable contribution, and that is also why I initially gave a positive rating. **However, the data insights of this paper differ from MIDAS.**
> > > > >
> > > > > MIDAS introduces 3D movie data, with depth obtained from **stereo matching by the authors**, which involves post-processing and is not of very high quality. In contrast, this paper appears to obtain depth from meshes (please correct me if I am mistaken). **Depth obtained from mesh rendering should be of very high quality, and this might be the key factor in improving performance.** Therefore, if game data curation is considered a contribution in this paper, it is acceptable not to release the data, but the authors should provide detailed information about how the depth maps were obtained. Additionally, the process should be legal.

---

> > > > > > ### Author Response · Authors · 2024-11-23
> > > > > > **Author Response**
> > > > > >
> > > > > > Thank you for your insightful comments and positive feedback on our manuscript. We appreciate your interest in understanding the depth acquisition process, as well as your recognition of the value that large-scale game data brings to depth learning. We will provide a detailed description of the game data curation process in the appendix as follows:
> > > > > >
> > > > > > **Understanding Modern Game Rendering Pipelines:**
> > > > > >
> > > > > > In contemporary game engines, the rendering process is central to transforming geometric data into visual output. This involves several key components and techniques:
> > > > > >
> > > > > > 1). Vertex and Geometry Processing: Geometric data is first processed to determine the position and attributes of each vertex in 3D space. This includes applying transformations based on camera perspective.
> > > > > >
> > > > > > 2). Rasterization: The processed geometric data is then converted into pixels in a process known as rasterization. This stage determines which pixels on the screen correspond to parts of the 3D models, establishing depth and color data for each pixel.
> > > > > >
> > > > > > 3). Shading: Modern shaders compute the lighting, coloring, and textures. Deferred shading, a popular technique in current game engines, separates lighting calculations from geometry rendering to optimize processing. Buffers such as depth and normal buffers store intermediate data essential for accurate lighting and shading. The depth buffer specifically stores depth information for each pixel, representing the distance from the camera to visible surfaces.
> > > > > >
> > > > > > **Accessing the Depth Buffer Using a Tool:**
> > > > > >
> > > > > > With an understanding of the modern game rendering pipeline, we can utilize an open-source tool, such as RenderDoc, to capture and analyze the depth buffer. Here's how the tool typically achieves this:
> > > > > >
> > > > > > 1). API Hooking and Integration: The tool hooks into a graphics API (such as DirectX or OpenGL) by integrating itself at runtime. It intercepts key API functions responsible for rendering operations, such as attaching and clearing depth buffers. In OpenGL, functions like `glFramebufferTexture2D` (for depth texture attachment), `glClear` with `GL_DEPTH_BUFFER_BIT` (for depth buffer clearing), and `glDrawElements` (for rendering operations that write to the depth buffer) are monitored to track the usage of depth buffers.
> > > > > >
> > > > > > 2). Capturing Depth Buffer Data: During rendering, the tool captures depth buffer data by recording how the depth information is initialized, updated, and utilized across frames. This enables the preservation of depth buffer contents for further analysis and usage.
> > > > > >
> > > > > > Since our data collection process is similar to that described in [1], we believe that our data acquisition process is legal.

---

> > > > > > > ### Comment · Reviewer_ATwr · 2024-11-24
> > > > > > >
> > > > > > > Thanks, authors and reviewer ```gu5t``` for having the discussions and providing more information on the data extraction part. However, I am not fully convinced by the authors' responses. **Still, there is a concern on the copyright issue, and it can affect the paper's contribution.**
> > > > > > >
> > > > > > > ---
> > > > > > >
> > > > > > > One of the main contributions of the paper is about the video depth data extracted from the game engine. It's clearly stated in the abstract and the introduction in the paper.
> > > > > > >
> > > > > > >
> > > > > > > > (related to MiDaS)"...we believe that the openness of data should not be the sole criterion for evaluating the quality or impact of a paper..."
> > > > > > >
> > > > > > > Of course, it should not be the sole criterion, but as long as it's one of the main contributions, the dataset part should be clear, transparent, and without any potential legal issues.
> > > > > > > Also, MiDaS didn't say that open-sourcing 3D movie training data is their contribution. MiDaS didn't release the data due to the copyright issue. As far as I understood correctly, their main contribution is the first zero-shot model that generalizes well to any domain.
> > > > > > >
> > > > > > >  On the other hand, the DA-V dataset is claimed as a main contribution and promised to be released. If the DA-V dataset cannot be released or cannot be used for future research, it diminishes the contribution (although the results are nice). That's why we are trying to make it clear in the review stage.
> > > > > > >
> > > > > > >
> > > > > > > > "...our data collection process is similar to that described in [1], we believe that our data acquisition process is legal..."
> > > > > > >
> > > > > > > Maybe, but I respectfully disagree. The point of the discussion is not whether any illegal tools are used or not. Also, "previous work did it" doesn't mean that others can do it too. Each game product can have its own terms of use.
> > > > > > >
> > > > > > > Buying their product does not mean that it's allowed to fully exploit their asset, transform it for our own use, and redistribute it for other uses. **I think it's recommended to directly contact the game companies and get permission for the extracted data for research use and open-sourcing**. I am not sure if consumers by themselves can determine if it's legal or not.
> > > > > > >
> > > > > > >
> > > > > > > ---
> > > > > > >
> > > > > > > Also, the gain from the usage of the proposed DA-V dataset is marginal. According to [the ablation study](https://openreview.net/forum?id=gWqFbnKsqR&noteId=JibYiFc0dr), the gain is 0.78% on average (max. 1.5% on KITTI) although it uses a significantly large amount of extra dataset (DA-V (6M) vs Baseline (0.093M)). I wonder why the gain is not so significant.

---

> > > > > > > > ### Comment · Reviewer_gu5t · 2024-11-26
> > > > > > > >
> > > > > > > > Thank you to the authors for providing additional details and to Reviewer ATwr for the valuable discussion.
> > > > > > > >
> > > > > > > > I agree with Reviewer ATwr that the capabilities demonstrated in previous work do not automatically validate the claims made in this paper. Moreover, the copyright concerns raised regarding this paper remain unaddressed. If this paper relies on RenderDoc, as referenced in [1], the issue becomes even more problematic.
> > > > > > > >
> > > > > > > > As stated in the [RenderDoc Documentation](https://github.com/baldurk/renderdoc):
> > > > > > > > "RenderDoc is intended for debugging your own programs only. Any discussion of capturing programs that you did not create will not be allowed in any official public RenderDoc setting, including the issue tracker, discord, or via email. For example this includes **capturing commercial games that you did not create**, or capturing Google Maps or Google Earth."
> > > > > > > >
> > > > > > > > If the work in this paper involves capturing data from commercial games, it may potentially violate these terms, raising questions about its legality.

---

> > > > > > > > > ### Author Response · Authors · 2024-12-03
> > > > > > > > > **Author Response**
> > > > > > > > >
> > > > > > > > > Thank you for the detailed review and valuable feedback.
> > > > > > > > >
> > > > > > > > > We have legally purchased the relevant games used in our dataset and obtained permission from the respective game companies, allowing the use of game content for personal, non-commercial educational purposes.
> > > > > > > > > For example, Take-Two Interactive replied:
> > > > > > > > >
> > > > > > > > > >Take-Two Interactive does not object to our fans using materials for non-commercial uses in a manner which does not intentionally spoil the plot for others.
> > > > > > > > >
> > > > > > > > > Similarly, Blizzard Entertainment supports the use of its game assets for educational purposes, stating:
> > > > > > > > >
> > > > > > > > > >Blizzard Entertainment supports the use of its game assets for educational purposes, and you are welcome and encouraged to create a Production for a school project, master's thesis, etc.
> > > > > > > > >
> > > > > > > > > Up to now, we have explicitly obtained permission from the following game companies to share their game content solely for educational purposes: *Cyberpunk 2077*, *Paragon*, *GTAV*, *The Witcher 3*, *Infinity Blade*, *Red Dead Redemption II*, and *Cities: Skylines*.
> > > > > > > > > Together, the number of frames from these games accounts for a significant portion of the DA-V dataset.
> > > > > > > > > We are still actively working to obtain permission from other game companies to expand the accessibility of our dataset.
> > > > > > > > >
> > > > > > > > > While we are authorized to share the existing content, we must adhere to certain restrictions. For instance, no spoilers are allowed to ensure that no key plot details or narrative elements are revealed. Additionally, the content is strictly limited to non-commercial use.
> > > > > > > > > As this involves more detailed terms, we do not elaborate on them in OpenReview.
> > > > > > > > > We are actively addressing these restrictions and aim to release the dataset as soon as possible, ensuring full compliance with all legal and ethical requirements.
> > > > > > > > >
> > > > > > > > > In addition to the dataset, we would like to emphasize the following contributions:
> > > > > > > > >
> > > > > > > > > 1). We demonstrate a scalable pipeline for generating depth data from diverse game environments and verify that high-quality synthetic data improves the generalization of video depth models in real-world scenarios.
> > > > > > > > >
> > > > > > > > > 2). We introduce the first method to produce consistent, high-resolution video depth predictions over 150 frames—15× longer than ChronoDepth—using a depth interpolation module, with efficient mixed-duration training that trains on short sequences while generalizing to longer videos with varying frame rates and resolutions.
> > > > > > > > >
> > > > > > > > > 3). Our model sets a new benchmark for accuracy and robustness in video depth estimation. The code and model weights are publicly available to support further research, similar to the practice of releasing only model weights in large language models.

---

### Meta-Review · Area_Chair_naM6 · 2024-12-14

**Metareview:**

The paper proposes an approach to video depth estimation with a generative model, integrating prior RoPE and flow matching, introducing a strategy for training on video clips of varying lengths, and further proposing a video depth interpolation technique for inference at high-resolution. The paper also describes a data synthesis pipeline to capture video depth ground truth from commercial video games.
It leverages the pipeline to generate a dataset with 40K video clips that is further used to train the proposed model.
The method outperforms prior generative models in spatial accuracy and temporal consistency as demonstrated through experimental results.

Reviewers appreciated the strong empirical results and found claims sufficiently validated through controlled experiments.
The paper is well written and easy to follow.

Reviewers requested additional information on the implementation details, and data synthesis pipeline. These were sufficiently met during the discussion phase. Concerns about insufficient improvements in metrics due to the proposed dataset as pointed out by reviewer ATwr are not shared by the AC - the supposedly small improvements in the delta1 metric would correspond to a ~20% reduction in error, which are significant. The main concern of the reviewers is the legality of the data collection as the proposed method
extracts internal representations from commercial games leveraging open source debugging tools.
This type of data extraction, the use and distribution may conflict with end user license agreements and copyrights.
Notably, prior work has set precedence for this kind of data collection and the authors obtained permission from a subset of concerned game publishers to release the data under specific conditions, which were not fully provided during the discussion period.

The AC considers the contribution and demonstrated experimental results as sufficient for acceptance, but nevertheless agrees with the request for an ethics review by the reviewers. As the dataset is a major contribution of the paper, its release and legality for follow-up research will strongly affect the impact of this paper.

**Additional Comments On Reviewer Discussion:**

Request regarding details on data collection, denoising steps, insights on the performance of 2D VAEs vs 3D VAEs, and further implementation details were met. These additional details should be incorporated into the paper or supplemental material.
The majority of the discussion evolved around the legality of the data collection and redistribution of the training data. Following precedence from prior work and the already obtained permissions from a subset of game publishers, it can be assumed that at least part of the dataset will be released under terms that allow non-commercial use for research purposes, and thus would benefit the research community.

---

### Decision · Program_Chairs · 2025-01-22

Accept (Poster)